# Heme biosynthesis depends on previously unrecognized acquisition of iron-sulfur cofactors in human amino-levulinic acid dehydratase

Gang Liu[1], Debangsu Sil [2], Nunziata Maio [1], Wing-Hang Tong [1], J. Martin Bollinger Jr.[2,3], Carsten Krebs [2,3✉] & Tracey Ann Rouault [1✉]

Heme biosynthesis and iron-sulfur cluster (ISC) biogenesis are two major mammalian metabolic pathways that require iron. It has long been known that these two pathways interconnect, but the previously described interactions do not fully explain why heme biosynthesis depends on intact ISC biogenesis. Herein we identify a previously unrecognized connection between these two pathways through our discovery that human aminolevulinic acid dehydratase (ALAD), which catalyzes the second step of heme biosynthesis, is an Fe-S protein. We find that several highly conserved cysteines and an Ala306-Phe307-Arg308 motif of human ALAD are important for $[Fe_4S_4]$ cluster acquisition and coordination. The enzymatic activity of human ALAD is greatly reduced upon loss of its Fe-S cluster, which results in reduced heme biosynthesis in human cells. As ALAD provides an early Fe-S-dependent checkpoint in the heme biosynthetic pathway, our findings help explain why heme biosynthesis depends on intact ISC biogenesis.

[1] Eunice Kennedy Shriver National Institute of Child Health and Human Development, National Institutes of Health, Bethesda, MD, USA. [2] Department of Chemistry, The Pennsylvania State University, University Park, PA 16802, USA. [3] Department of Biochemistry and Molecular Biology, The Pennsylvania State University, University Park, PA 16802, USA. ✉email: cdk10@psu.edu; rouault@mail.nih.gov

In mammalian cells, substantial amounts of iron are consumed by heme biosynthesis and iron-sulfur cluster (ISC) biogenesis[1,2]. Heme is synthesized by eight sequential enzymatic steps[3], of which the first takes place in the mitochondrial matrix, where 5-aminolevulinate synthase (ALAS) catalyzes the condensation of succinyl-CoA with glycine to generate ALA (Supplementary Fig. 1)[3,4]. Two *ALAS* genes are present in vertebrates, a ubiquitously expressed *ALAS1* and an erythroid-specific *ALAS2*[2,5]. The mRNA of *ALAS2* has an iron responsive element (IRE) in its 5'-untranslated region (UTR) and is post-transcriptionally regulated by the iron regulatory proteins (IRP1 and IRP2)[6,7]. Since IRP1 loses its IRE-binding activity when it ligates an Fe-S cluster, ALAS2 protein levels are indirectly regulated by ISC biogenesis because (i) binding of apo-IRP1 to the IRE represses ALAS2 translation and (ii) holo-IRP1 (with its Fe-S cluster intact) lacks IRE-binding activity[6,7]. ALAS1, which catalyzes the rate-limiting step of heme biosynthesis in non-erythroid cells[3], is subject to negative feedback regulation by cellular heme content[8,9]. After its synthesis in mitochondria, ALA is exported to the cytosol, where ALA dehydratase (ALAD) catalyzes the second step of heme biosynthesis by condensing two ALA molecules into porphobilinogen (Supplementary Fig. 1). ALAD is evolutionarily conserved and constitutes an important enzyme for biosynthesis of chlorophyll, the corrin ring of vitamin B12, and other important tetrapyrroles[10,11]. After three additional enzymatic reactions in the cytosol and two in the mitochondria, the final insertion of ferrous iron into protoporphyrin IX to generate heme occurs in the mitochondrial matrix and is catalyzed by ferrochelatase (FECH) (Supplementary Fig. 1)[4]. Human FECH is an Fe-S protein[12–14], and its stability depends on coordination of a $[Fe_2S_2]$ cluster[12,15], whereas the FECH of *S. cerevisiae* is not an Fe-S protein[12,14].

Iron-sulfur clusters are ancient prosthetic groups with essential biological functions[16–19]. In the mitochondria of mammalian cells, ISCs are assembled de novo by a complex composed of NFS1, ISD11, ACP, the ISCU scaffold and frataxin[20]. After assembly, nascent clusters are transferred by the HSPA9/HSC20 chaperone/cochaperone system directly to recipient proteins or through intermediate scaffolds[21]. Previous models have proposed that initial Fe-S synthesis occurs solely in the mitochondrial matrix[22]. However, the core mammalian ISC components have also been identified in the cytosol and nucleus, and accumulating evidence shows that the ISC biogenesis machineries likely operate independently to generate nascent clusters in several subcellular compartments of multicellular eukaryotes[23–25].

Thus far, there are two well-characterized nodes in the heme biosynthetic pathway of mammalian cells at which defects in the Fe-S biogenesis machinery can suppress heme synthesis. First, FECH of higher eukaryotes contains a $[Fe_2S_2]$ cluster that is proposed to stabilize the enzyme[12,13]. Second, ALAS2 is post-transcriptionally regulated in erythroid cells by the interconversion of IRP1 between its holo-form, which functions as cytosolic aconitase, and its IRE-binding apo form, which represses ALAS2 expression[7]. Therefore, Fe-S biogenesis defects can block heme synthesis by either repressing ALAS2 synthesis in erythroid cells or inactivating FECH. However, it remains unexplained how Fe-S biogenesis defects in yeast result in impaired heme production[26,27], given that yeast FECH is not an Fe-S protein and yeast lack IRPs[12,14]. The absence of an obvious explanation for the observed link between Fe-S biogenesis and heme biosynthesis in yeast drove us to search for unrecognized intersections between the Fe-S and heme biosynthesis pathways.

We previously reported that binding of HSC20 to a leucine-tyrosine-arginine (LYR) motif of succinate dehydrogenase complex subunit B (SDHB) was essential for ISC incorporation into SDHB[28]. The LYR motif was also identified in other HSC20-binding proteins[28,29]. Therefore, we speculated that analyzing protein sequences for the presence of the LYR motif and motifs with similar chemical properties (LYR-like motifs) could be used to discover candidate Fe-S proteins[28,30]. Although the importance of the LYR motif for ISC acquisition was initially discovered in studies of SDHB, in which mutagenesis of LYR-like motifs in the sequences of either SDHB[28] itself or in the accessory factor required for SDH assembly, known as SDHAF1[31], impaired ISC transfer to SDHB. Subsequent experiments demonstrated that LYR motifs were also important for ISC acquisition in respiratory chain complexes I and III as they were able to mediate direct binding of the cochaperone HSC20 (HSCB) that facilitated ISC transfer from the primary Fe-S biogenesis complex[29]. Short motifs are compact binding modules that typically regulate and coordinate protein processing, localization and degradation events[32], often by engaging in low affinity transient interactions with a binding site in a protein domain[32].

Our previous extensive studies defined some molecular requirements for a three-residue amino acid sequence to function as a LYR-like motif. The first position must be a small amino acid with a more aliphatic character, including leucine, isoleucine, valine and alanine. The second position must be either Y or F (aromatic), and the third position must be either R or K (positively charged)[28,30]. Therefore, we used bioinformatics to screen heme biosynthetic enzymes for the presence of LYR-like motifs, as they have been experimentally shown to bind the cochaperone HSC20 to facilitate ISC transfer from the initial biogenesis complex to recipient proteins. We found that ALAD contained a LYR-like motif ($A_{306}F_{307}R_{308}$). Our identification of a motif in human ALAD ($A_{306}F_{307}R_{308}$) that conforms to these rules, along with the presence of zinc in the crystal structure of the enzyme purified aerobically, suggested that ALAD might bind a Fe-S cluster.

Here, we show that ALAD does indeed require a $[Fe_4S_4]$ cofactor for optimal enzymatic activity upon overexpression in human cells and in bacteria. We also report additional functional studies to elucidate the molecular mechanisms by which human ALAD acquires and coordinates the $[Fe_4S_4]$ cluster, and to demonstrate the biological significance of the $[Fe_4S_4]$ cluster.

## Results

**Human ALAD coordinates a $[Fe_4S_4]$ cluster**. At the outset, we explored whether there might be as-yet-uncharacterized intersections between the Fe-S and heme biosynthetic pathways. We first simultaneously knocked down the mitochondrial and cytosolic ISCU isoforms, which constitute the primary Fe-S biogenesis scaffolds in mammalian cells. Overexpression of yeast FECH, a non-Fe-S enzyme, did not restore heme biosynthesis in ISCU-deficient cells (Supplementary Fig. 2a–c and Supplementary Results and Discussion). We next overexpressed a cytosolic ISCU mutant that impairs Fe-S biogenesis in the cytosol of mammalian cells and found that heme biosynthesis was inhibited (detailed in the Supplementary Results and Discussion and Supplementary Fig. 2d–f). Taken collectively, these results implied that one or more cytosolic heme biosynthetic steps were dependent on Fe-S biogenesis.

Then we screened the amino acid (AA) sequences of cytosolic heme biosynthesis enzymes for conserved LYR or LYR-like motifs that might indicate the presence of an Fe-S cluster ligated by the protein. Sequence alignment showed that human ALAD contains six cysteines, C119, C122, C124, C132, C162, and C223, and a LYR-like motif (Ala306-Phe307-Arg308) that are conserved among eukaryotic ALAD orthologs (Supplementary Fig. 3). Therefore, in terms of AA sequence and composition, ALAD

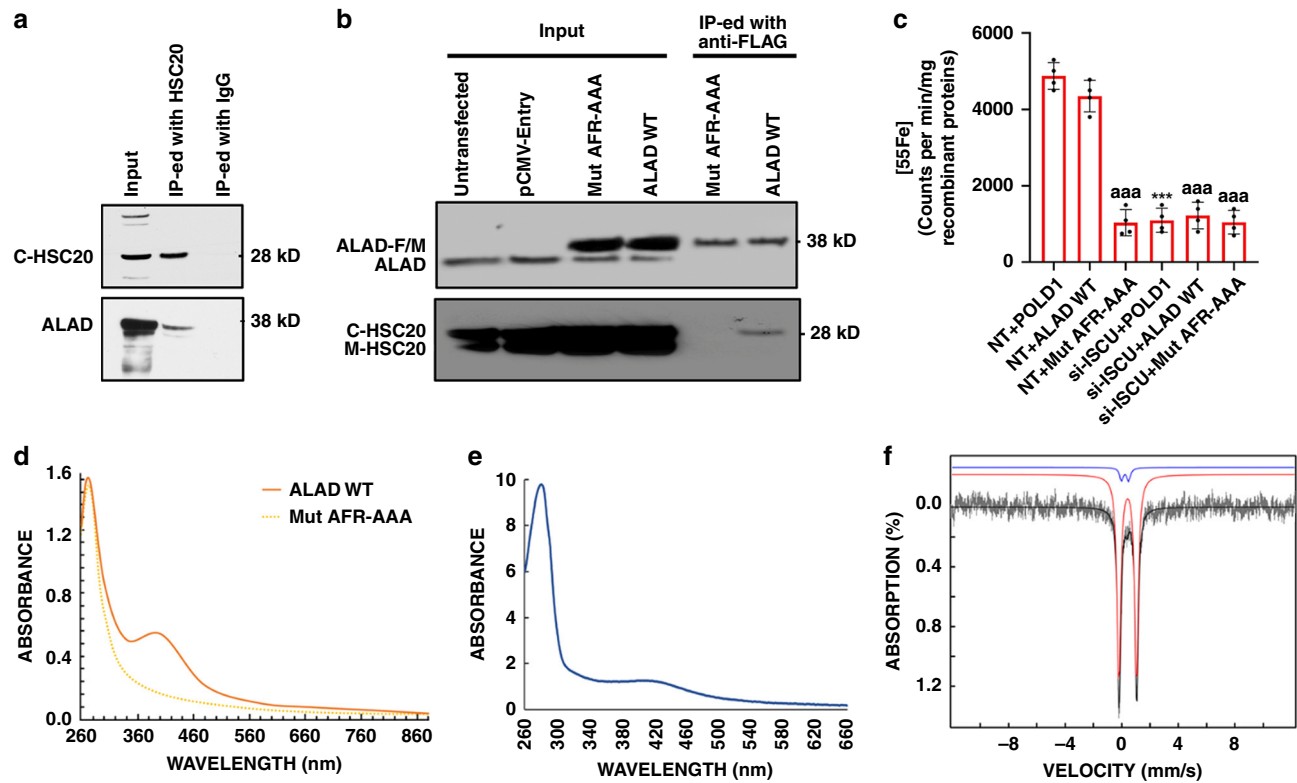

**Fig. 1 Human ALAD coordinates a [Fe₄S4] cluster. a** Immunoprecipitation (IP) of cytosolic fractions of HeLa cells using anti-HSC20 antibody shows that ALAD co-precipitates with HSC20. The input was 5% of the cytosolic fraction lysate, and the remaining 95% of the lysate was subjected to IP with anti-HSC20 antibody. The eluates were then subjected to Western blotting with anti-HSC20 and anti-ALAD antibodies. The experiments were repeated twice independently with similar results **b** IP of whole-cell lysates of HeLa cells transfected with plasmids expressing C-terminally FLAG/MYC-tagged wild-type ALAD (ALAD WT) or the AFR to AAA mutant (Mut AFR-AAA). The inputs were 10% of lysates used in the IPs. Untransfected HeLa cells or HeLa cells transfected with empty vector (pCMV-Entry) were used as controls. The experiments were repeated twice independently with similar results. These results indicated that the AFR motif of human ALAD was important for its interaction with HSC20. **c** $^{55}$Fe incorporation into C-terminally FLAG/MYC-tagged POLD1 (POLD1), ALAD WT and Mut AFR-AAA in HEK293 cells transfected with nontargeting siRNA (NT) or On-TARGETPlus siRNA pools against human *ISCU* (si-ISCU). The data are presented as mean values ± standard deviations of $n = 4$ biologically independent samples. Two-sided unpaired Student's *t*-test was employed to analyze statistical significance. No multiple comparison was performed. ***$P < 0.001$ versus the "NT + POLD1" group; $^{aaa}P < 0.001$ versus the "NT + ALAD WT" group. The exact *P* values are provided in the Source Data file. **d** UV-vis spectra of C-terminally FLAG/MYC-tagged ALAD WT (orange line) and Mut AFR-AAA (yellow line) that were expressed in Expi293 cells and purified anaerobically. **e** UV-vis spectrum of the MBP-ALAD purified anaerobically from *E. coli* showing absorbance at 420 nm. **f** Experimental 4.2-K/53-mT Mössbauer spectrum of the MBP-ALAD purified anaerobically from *E. coli* (black vertical bars) and simulation (black solid line) along with contributions from individual subspectra associated with [Fe₄S₄]²⁺ (red line) and [Fe₂S₂]²⁺ clusters (blue line) with parameters quoted in the text.

appeared to be a candidate Fe-S protein that might link ISC biogenesis and heme biosynthesis.

As we have previously shown that de novo Fe-S biosynthesis occurs in parallel in the mitochondrial and cytosolic compartments of mammalian cells[23], we first determined whether human ALAD interacted with cytosolic HSC20. When the cytosolic fraction of HeLa cells was immunoprecipitated with anti-HSC20 antibody, the eluate contained ALAD (Fig. 1a), suggesting that endogenous cytosolic HSC20 (C-HSC20) interacted with ALAD. Because HSC20 binds directly to LYR motifs, permitting transfer of nascent clusters from ISCU to recipient proteins[16,28], we then determined whether the AFR motif of human ALAD was important for its interaction with HSC20. We transfected HeLa cells with plasmids expressing FLAG/MYC-tagged wild-type ALAD (ALAD WT) or the ALAD AFR to alanine mutant (Mut AFR-AAA). Lysates from these cells were immunoprecipitated with anti-FLAG antibody, and the eluates were analyzed by western blotting. As shown in Fig. 1b, wild-type ALAD interacted specifically with C-HSC20 but not with the lower-molecular-weight mitochondrial HSC20 (M-HSC20), as expected, given that ALAD is a cytosolic enzyme. No interaction between C-HSC20

and the AFR-AAA mutant was detected. Collectively, these data demonstrated that ALAD interacted with C-HSC20 and that the AFR motif of ALAD was important for the interaction.

To investigate whether human ALAD coordinates an ISC in human cells, we quantified $^{55}$Fe incorporation into C-terminally FLAG/MYC-tagged ALAD WT, C-terminally FLAG/MYC-tagged Mut AFR-AAA and C-terminally FLAG/MYC-tagged POLD1 (a protein known to acquire its [Fe₄S₄] cluster in the cytosol[33]) in HEK293 cells transfected with either a pool of nontargeting siRNAs (NT) or with siRNAs directed against human *ISCU* (si-ISCU). We found that ALAD and POLD1 bound comparable amounts of $^{55}$Fe in control HEK293 cells transfected with nontargeting siRNAs (Fig. 1c and Supplementary Fig. 4a). Mut AFR-AAA, which did not interact with C-HSC20 (Fig. 1b), bound significantly less $^{55}$Fe than ALAD WT in NT HEK293 cells (Fig. 1c, $P < 0.001$). Compared with control cells, ALAD and POLD1 bound significantly less $^{55}$Fe in HEK293 cells with their ISC assembly machinery rendered defective by silencing of human *ISCU* (Fig. 1c and Supplementary Fig. 4a, $P < 0.001$). Taken together, these results demonstrated that ALAD bound iron, presumably in the form of an ISC, in human cells. To

determine whether human ALAD coordinated an ISC in human cells, we overexpressed C-terminally FLAG/MYC-tagged ALAD WT and Mut AFR-AAA in Expi293 cells and purified the recombinant proteins anoxically. Anaerobically purified recombinant ALAD migrated as a single band on SDS-PAGE (Supplementary Fig. 4b) and exhibited a shoulder at ~420 nm in its UV-vis absorption spectrum (Fig. 1d), suggesting the presence of $[Fe_4S_4]$ cluster(s)[34–36].

We next expressed N-terminally MBP-tagged human ALAD (MBP-ALAD) in growing bacteria along with the *A. vinelandii isc* operon, which encodes the basic Fe-S biogenesis proteins and can assist iron-sulfur cluster assembly in bacteria, and purified the expressed protein anoxically[37,38]. The anaerobically purified MBP-ALAD was brown (Supplementary Fig. 4c), migrated as a single band on SDS-PAGE (Supplementary Fig. 4d), and also exhibited a shoulder at ~420 nm in its UV-vis absorption spectrum (Fig. 1e).

To validate the presence of (an) $[Fe_4S_4]$ cluster(s) and determine cluster stoichiometry, a $^{57}$Fe-enriched MBP-ALAD sample was analyzed by Mössbauer and EPR spectroscopies. The 4.2-K Mössbauer spectrum collected in a 53 mT magnetic field applied parallel to the direction of γ radiation (Fig. 1f, black vertical bars) was dominated by a quadrupole doublet with parameters typical of $[Fe_4S_4]^{2+}$ clusters [isomer shift (δ) of 0.44 mm/s and quadrupole splitting parameter ($\Delta E_Q$) of 1.24 mm/s, 95% of total intensity, red line][17,39]. In addition, the spectrum revealed a second minor quadrupole doublet with parameters typical of an $[Fe_2S_2]^{2+}$ cluster (δ = 0.28 mm/s and $\Delta E_Q$ = 0.50 mm/s, 5% of total intensity, blue line). As wild-type ALAD binds 3.5 Fe per protein monomer (vide infra), the Mössbauer results implied that each ALAD monomer contained ~0.8 $[Fe_4S_4]^{2+}$ and ~0.1 $[Fe_2S_2]^{2+}$. $[Fe_2S_2]^{2+}$ clusters are known to form by degradation of $[Fe_4S_4]^{2+}$ clusters[37]. Therefore, the small fraction of $[Fe_2S_2]^{2+}$ cluster associated with the minor doublet could have arisen by breakdown of a fraction of the $[Fe_4S_4]^{2+}$ clusters. Consistent with the Mössbauer analysis, the EPR spectrum of as-purified MBP-ALAD showed no signal (Supplementary Fig. 5a), while that of the dithionite-reduced MBP-ALAD showed weak signals corresponding to $[Fe_4S_4]^+$ cluster resulting from the reduction of the $[Fe_4S_4]^{2+}$ cluster (Supplementary Fig. 5b). Taken together, these results demonstrated that human ALAD harbored one $[Fe_4S_4]$ cluster per monomer when it was expressed in *E. coli* along with the *isc* operon and purified anaerobically.

It is known for other enzymes that employ $[Fe_4S_4]$ cofactors to promote dehydration reactions (e.g., dihydroxyacid dehydratase, aconitase, quinolinate synthase) that the substrate is coordinated by the unique Fe site of the cluster that lacks a cysteine ligand from the protein[40,41]. Substrate coordination by the three-coordinate iron of the cluster can perturb the Mössbauer spectra of the $[Fe_4S_4]^{2+}$ cofactors and EPR spectra of the $[Fe_4S_4]^+$ forms[42]. We therefore tested whether binding of the substrate, ALA, or of the potent ALAD inhibitor, succinylacetone (SA)[43], to the $[Fe_4S_4]$-containing recombinant ALAD perturbs the spectroscopic features of the cofactor. Addition of either the substrate or the inhibitor did not cause a detectable perturbation to either the Mössbauer spectrum of the oxidized form of the cluster or the EPR spectrum of the dithionite-reduced protein (Supplementary Fig. 6). These observations suggested either that the substrate/inhibitor/product did not coordinate the cluster or that their coordination did not perturb the spectra. Alternatively, the octameric nature of the protein and fraction-of-sites behavior[44] could result in a lower fraction of substrate/inhibitor-bound form, making any perturbation more difficult to detect.

**The $[Fe_4S_4]$ cluster is important for ALAD activity in vitro.** ALAD was previously identified as a zinc enzyme, and structures with zinc in the active site have been studied[45,46]. Therefore, to

determine whether the $[Fe_4S_4]$ cluster detected in human ALAD is functionally significant, we compared the enzymatic activities of aerobically purified wild-type apo-ALAD (aero-apo-ALAD WT, see the Methods section for details), aerobically purified wild-type ALAD (aero-ALAD WT, aerobically purified wild-type ALAD prior to treating with EDTA), aerobically purified wild-type zinc-bound-ALAD (aero-Zn-ALAD WT), and the anaerobically purified wild-type ALAD. As demonstrated by the ICP-MS analysis, aero-apo-ALAD WT did not contain iron or zinc, while aero-ALAD WT contained no zinc and a small amount of iron (Fig. 2a). The enzymatic activity of aero-Zn-ALAD WT was approximately six times higher than aero-apo-ALAD WT and two times higher than aero-ALAD WT (Fig. 2a, b, $P < 0.001$). These results suggested that at least partial activity was conferred by bound zinc. Wild-type ALAD expressed along with *isc* operon and purified anaerobically (ALAD WT) did not contain zinc (Fig. 2a). Its enzymatic activity was approximately three times higher than that of aero-Zn-ALAD WT (Fig. 2b, compare the fourth bar versus the third bar). These results demonstrated that $[Fe_4S_4]$-ALAD had significantly greater enzymatic activity than Zn-ALAD.

To identify the AAs important for the coordination/acquisition of the $[Fe_4S_4]$ cluster, we substituted each of the six highly conserved cysteines, and Lys252, which has been shown to form a Schiff base with the carbonyl of one of the two ALA substrates in the enzymatic active site (Supplementary Fig. 7)[47]. Additionally, we substituted the LYR-like motif of human ALAD with alanines. Thus, eight MBP-ALAD mutants (C119A, C122A, C124A, C132A, C162A, C223A, K252M, and Mut AFR-AAA) were generated. We expressed these mutants along with the bacterial *isc* operon and purified the overexpressed proteins anaerobically. To investigate whether the mutants coordinated a $[Fe_4S_4]$ cluster, we plotted their molar extinction coefficients based on their UV-vis absorption spectra. As shown in Fig. 2c, none of UV-vis spectra of the mutants exhibited the absorption at ~420 nm, the hallmark feature of $[Fe_4S_4]$ clusters. This observation indicated that these AAs were involved in either $[Fe_4S_4]$ cluster coordination/acquisition or in maintaining the ALAD protein in a structure conducive to $[Fe_4S_4]$ cluster coordination/acquisition. Importantly, the ALAD mutants all had significantly less iron and lower (or undetectable) enzymatic activities than ALAD WT (Fig. 2a, b, all $P$ values < 0.01). These results further confirmed that the cysteines and AFR motif were important for $[Fe_4S_4]$ cluster coordination/acquisition and the $[Fe_4S_4]$ cluster was important for human ALAD enzymatic activity in vitro.

To determine whether the N-terminal MBP tag hindered the formation of a homo-octamer of ALAD subunits that constitutes the previously defined active form of ALAD[10], and to assess how $[Fe_4S_4]$ cluster coordination affected ALAD quaternary structure, the anaerobically purified wild-type and mutant ALAD proteins were subjected to native PAGE electrophoresis. As shown in Fig. 2d and Supplementary Fig. 8, most of the ALAD WT and aero-zinc-ALAD WT monomers formed ~640 kD homo-octamers, while all the ALAD mutants and the aero-apo-ALAD WT mostly assembled into ~480 kD homo-hexamers, a previously identified less active multimeric state[10]. These results suggested that MBP-ALAD can form homo-octamers in vitro and that $[Fe_4S_4]$ cluster coordination correlated with formation of higher proportions of ALAD homo-octamers.

**The $[Fe_4S_4]$ cluster is important for ALAD activity in cells.** Next, we investigated the biological significance of the $[Fe_4S_4]$ cluster of ALAD in human HepG2 cells. We cotransfected HepG2 cells with siRNAs to knockdown (KD) the expression of endogenous ALAD, and used a plasmid to direct expression of either recombinant wild-type ALAD (the ALAD WT group) or the ALAD mutant in

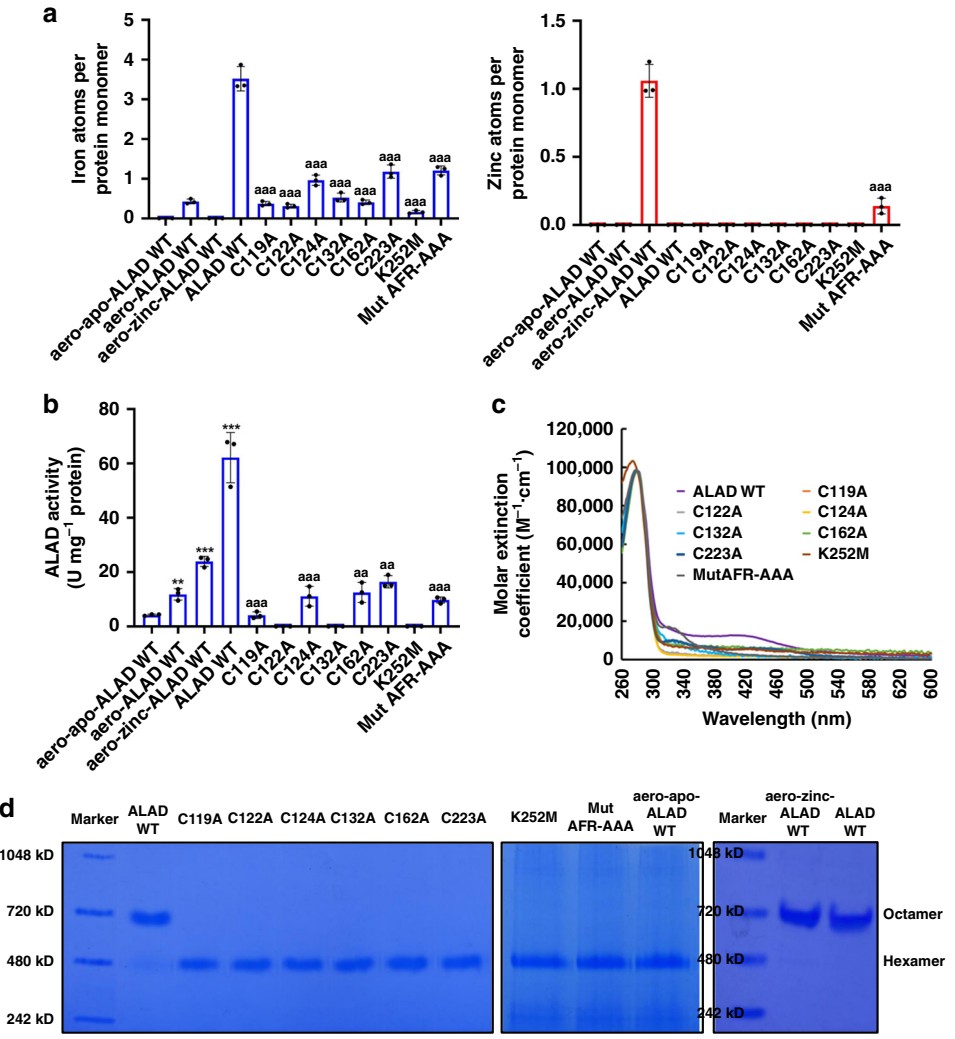

**Fig. 2 The [Fe₄S₄] cluster is important for ALAD enzymatic activity in vitro. a** ICP-MS analysis and **b** Enzymatic activity of aerobically purified apo-ALAD (aero-apo-ALAD WT), aerobically purified zinc-ALAD (aero-Zn-ALAD WT) and the wild-type (ALAD WT) and mutant ALAD proteins expressed along with the *isc* operon in *E. coli* and purified anaerobically. Results indicate that the [Fe₄S₄]²⁺ cluster was important for the enzymatic activity of human ALAD in vitro. The data in panels **a** and **b** are presented as mean values ± standard deviations of n = 3 biologically independent samples. Two-sided unpaired Student's *t*-test was employed to analyze statistical significance. No multiple comparison was performed. **, *** represent *P* < 0.01 and *P* < 0.001, respectively, versus the "aero-apo-ALAD WT" group; aa, aaa represent *P* < 0.01 and *P* < 0.001, respectively, versus the "ALAD WT" group. The exact *P* values are provided in the Source Data file. **c** Molar extinction coefficients of C119A, C122A, C124A, C132A, C162A, C223A, K252M, and Mut AFR-AAA mutants that were expressed along with the *isc* operon in *E. coli* and purified anaerobically. The results indicated that C119, C122, C124, C132, C162, C223, and K252 of human ALAD were important for [Fe₄S₄] cluster coordination and/or retention, and the AFR motif was important for [Fe₄S₄] cluster acquisition. **d** Native gel electrophoresis analysis of wild-type and mutant ALAD proteins expressed along with the *isc* operon in *E. coli* and purified anaerobically, the aero-apo-ALAD WT protein, and aero-Zn-ALAD WT protein (See Supplementary Fig. 8 for the full Zn- ALAD WT native gel). This experiment was repeated three times independently with similar results. The results suggest that [Fe₄S₄] cluster coordination correlated with increased proportions of ALAD homo-octamers.

which the AFR motif had been replaced by alanines (Mut AFR-AAA group). Effective KD of endogenous ALAD was achieved in HepG2 cells (Fig. 3a). Recombinant ALAD WT or the Mut AFR-AAA mutant were well expressed with undetectable background expression of endogenous ALAD (Fig. 3a, b, *P* values < 0.01 compared with untransfected cells). We used ALAS1 protein levels and the activities of the heme-dependent enzymes, cytochrome P450 1A2 (CYP1A2) and catalase, as readouts of the integrity of the heme biosynthetic pathway[48–50]. As shown in Fig. 3a, d, e, respectively, there were no significant differences in the protein level of ALAS1 or in the catalase and CYP1A2 activities between the ALAD WT group and untransfected HepG2 cells. In cells expressing ALAD-Mut AFR-AAA, however, we observed significantly decreased ALAD, catalase and CYP1A2 activities in cell lysates as compared with either untransfected HepG2 cells or cells expressing ALAD WT (Fig. 3a,

c–e, *P* values < 0.001). These results demonstrated that the AFR-AAA mutant was nonfunctional and exhibited significantly lower enzymatic activity than wild-type ALAD. Therefore, overexpression of ALAD-Mut AFR-AAA did not restore heme biosynthesis that was suppressed by siRNA-induced ALAD deficiency. Given that the AFR motif of human ALAD is required for [Fe₄S₄] cluster acquisition, these results demonstrated that the [Fe₄S₄] of human ALAD was important for its enzymatic activity and heme biosynthesis in HepG2 cells.

Finally, we examined whether zinc addition affected ALAD enzymatic activity by adding zinc to the medium of mammalian cells overexpressing either WT ALAD or the Mut AFR-AAA (100 μM ZnCl₂ for 48 h). As shown in Fig. 3a, levels of human metallothionein MT2A, whose expression is regulated by zinc[51,52], were greatly increased, suggesting that levels of cellular

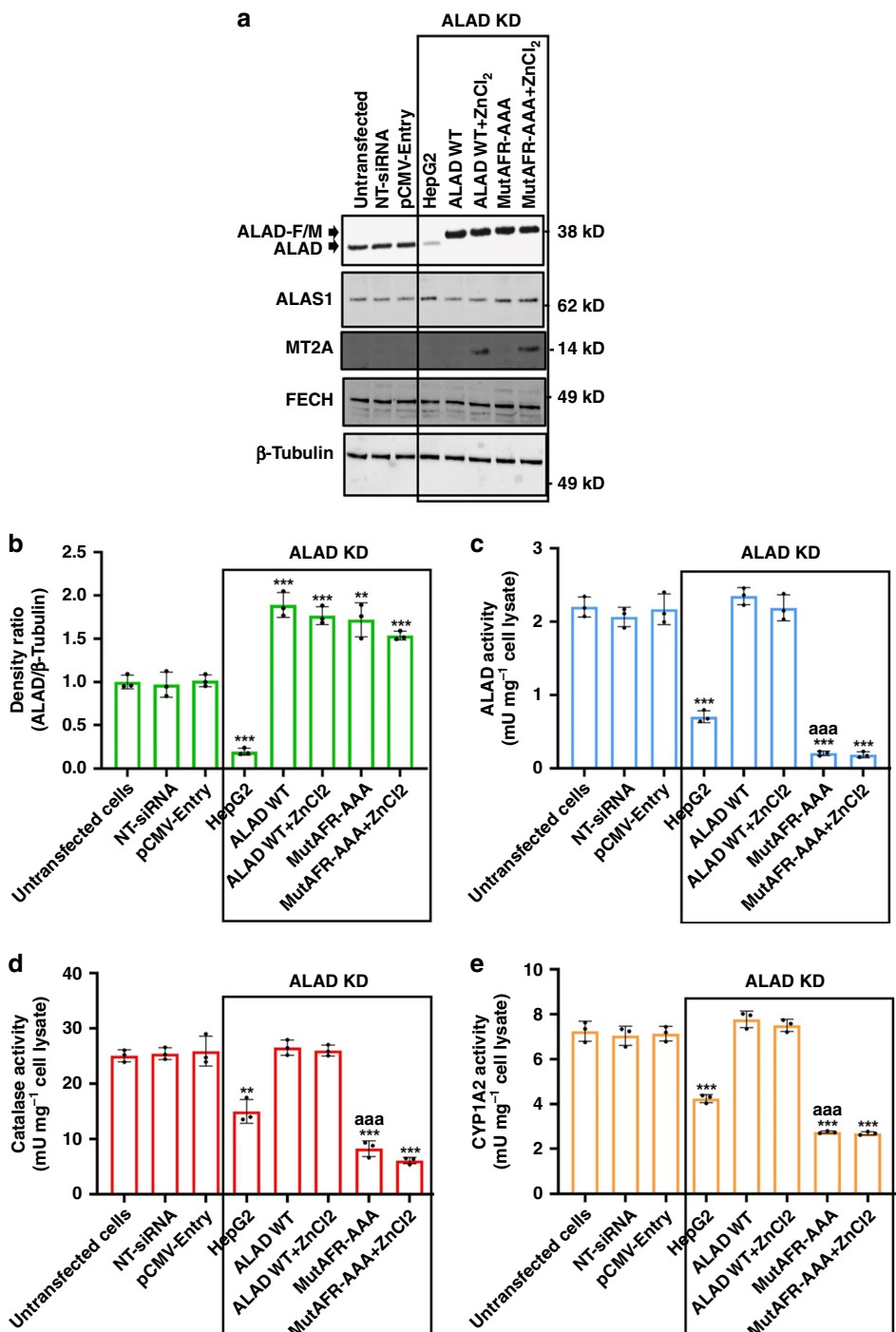

**Fig. 3 The [Fe$_4$S$_4$] cluster is important for ALAD activity in HepG2 cells. a** Western blot analysis of protein levels of endogenous and exogenous ALAD, ALAS1, MT2A, and FECH in untransfected HepG2 cells, HepG2 cells transfected with nontargeting siRNA (NT-siRNA) or pCMV-Entry empty vector (pCMV-Entry), ALAD deficient HepG2 cells (ALAD KD), HepG2 cells from the ALAD WT and Mut AFR-AAA groups without or with 100 µM ZnCl$_2$ addition (ALAD WT, ALAD WT + ZnCl$_2$, Mut AFR-AAA or Mut AFR-AAA + ZnCl$_2$, respectively). The experiments were repeated at least twice independently with similar results. **b** Densitometry of the ratios between the intensities of ALAD and the corresponding β-Tubulin protein levels. **c** ALAD, **d** catalase, and **e** CYP1A2 activities in lysates of these cells. The data in panels **b**, **c**, **d**, and **e** are presented as mean values ± standard deviations of $n = 3$ biologically independent samples. Two-sided unpaired Student's $t$-test was employed to analyze statistical significance. No multiple comparison was performed. **, *** represent $P < 0.01$ and $P < 0.001$, respectively, versus "untransfected cells"; aaa represents $P < 0.001$ versus the "ALAD WT" group. The exact $P$ values are provided in the Source Data file.

zinc indeed increased. However, there were no significant changes in either the protein levels of ALAS1 or the activity of ALAD, catalase or CYP1A2 after adding ZnCl$_2$ to the medium (Fig. 3a, c–e). These results indicated that increasing intracellular zinc concentrations did not increase ALAD enzymatic activity in

HepG2 cells overexpressing either wild-type ALAD or the AFR-AAA mutant. Notably, FECH protein levels were not affected by zinc addition, indicating that the coordination of [Fe$_2$S$_2$] cluster by human FECH was also unaffected by increased intracellular zinc levels (Fig. 3a).

## Discussion

In this study, we found that human ALAD requires a $[Fe_4S_4]$ cluster for full enzymatic activity both in vitro and in HepG2 cells. More importantly, the $[Fe_4S_4]$ cluster of ALAD is crucial for heme biosynthesis, unveiling a node of regulation of heme biosynthesis by ISC biogenesis that operates at the second step of the heme biosynthetic pathway.

To determine whether one or more steps of the cytosolic reactions of heme biosynthesis depended on Fe-S cluster biogenesis, we overexpressed a form of cytosolic ISCU that lacked a mitochondrial targeting signal and contained the mutation D46A, which prevents the ISCU scaffold from transferring its Fe-S cluster to recipient proteins in mammalian cells. Our results demonstrated that two heme-dependent enzymes, catalase and cytochrome P450 1A2, lost activity when the D46A ISCU construct was expressed to suppress cytosolic Fe-S biogenesis, as expected. We then used bioinformatics to evaluate whether the AA sequence of any of the cytosolic heme biosynthesis enzymes contained LYR-like motifs and potential cluster-ligating cysteines, which we have shown may represent features of candidate Fe-S proteins[30]. Only human ALAD was found to contain an LYR-like motif, consisting of Ala306-Phe307-Arg308. LYR-like motifs are not rare in the human proteome, and only a subset of these motifs bind the cochaperone HSC20 to acquire an ISC, perhaps because of spatial and temporal issues brought about by the multiple compartments of mammalian cells. ALAD is expressed in the cytosol of mammalian cells, in proximity to active protein complexes that actively synthesize ISCs de novo, and it was therefore considered a strong potential candidate[23]. Moreover, ALAD had been previously characterized as a zinc protein, as ALAD purified from beef liver contained much larger amounts of zinc than iron[53], and zinc can often substitute for ISC ligation in proteins that have been exposed to oxygen, permitting the ISC to degrade. The low enzymatic activity of metal-free ALAD could be partially restored by zinc addition[53,54], and in later studies, it was demonstrated that ALAD octamers required four zinc ions for enzymatic activity[45]. However, all these studies were performed under aerobic conditions that can interfere with identification of iron-sulfur clusters. In fact, many Fe-S proteins have been previously mischaracterized as zinc proteins, such as CPSF30[55], human CISD2[56], and yeast POL3[33]. Therefore, we decided to explore whether ALAD is an Fe-S protein. According to our results, human ALAD could transiently bind the cochaperone HSC20, which is dedicated to ISC biogenesis, and can coordinate an ISC in human cells. In addition, each human ALAD monomer harbored one $[Fe_4S_4]$ cluster when it was overexpressed in bacteria along with the isc operon and purified anaerobically.

We then compared the enzymatic activity of $[Fe_4S_4]$-ALAD with that of zinc-ALAD and found that $[Fe_4S_4]$-ALAD had significantly higher enzymatic activity than zinc-ALAD. ALAD catalyzes the asymmetric condensation of two ALA molecules to form porphobilinogen. The ALA molecule that becomes the propionyl side chain and pyrrole nitrogen-containing part of porphobilinogen is referred to as the P-side ALA, whereas that which contributes the acetyl side chain and amino nitrogen-containing part of porphobilinogen is known as the A-side ALA. According to previous studies, the first step of this enzymatic reaction is the formation of P-side Schiff base of ALA with the nitrogen of lysine 252, followed by A-side ALA binding to the other side of enzymatic active site pocket[44,54,57-59]. It was proposed that in zinc-ALAD, the zinc functioned as a Lewis acid to bind A-side ALA[58]. On the basis of the chemical similarity of the ALAD reaction to those catalyzed by the $[Fe_4S_4]$-containing enzymes aconitase[60], dihydroxyacid dehydratase[61], and quinolinate synthase[57,62], we initially posited that the $[Fe_4S_4]$ cluster of ALAD might be coordinated by only three Cys ligands,

permitting the non-Cys-coordinated iron to function as a Lewis acid and to coordinate the ALA substrate[60,63]. However, we found that neither the EPR nor the Mössbauer spectrum of $[Fe_4S_4]$-ALAD (at the concentration of 250 μM) was perturbed by the addition of either ~10 mM ALA or ~10 mM succinylacetone, a potent ALAD inhibitor (Supplementary Fig. 6). The absence of perturbation of the $[Fe_4S_4]$ cluster of ALAD upon addition of the substrate may be attributable to several unusual features of the enzyme. One is that the enzyme functions as an octamer, and it is not known how many monomeric subunits catalyze the enzymatic reaction at any given time. Interconnectedness between subunits implies that cooperativity and conformational changes are important[64]. Since $[Fe_4S_4]$-ALAD mainly assembled into octamers and each monomer coordinated one $[Fe_4S_4]$ cluster, it is conceivable that only a subset of the eight $[Fe_4S_4]$ clusters of octameric ALAD bound the ALA substrate, and substrate/product-coordinated forms of the enzyme failed to accumulate to a level detectable by Mössbauer spectroscopy. In fact, in the related enzyme, hydroxybutyryl-CoA dehydratase, there is no experimental evidence for coordination of the substrate to the non-cysteinyl-coordinated Fe site of its $[Fe_4S_4]$ cluster[65,66].

Interestingly, while the predominant oligomeric form of ALAD is that of a homo-octamer, a rare human ALAD mutation that leads to substitution of phenylalanine 12 into leucine (ALAD$^{F12L}$) has been reported to cause a profound rearrangement of the enzyme oligomeric form, with a switch to the hexameric form that exhibited only 12% of residual activity compared with the octameric enzyme[67,68]. The significantly lower activity of the hexameric enzyme compared with that of the octameric ALAD points to a potential cooperative mechanism of catalysis that warrants additional investigations for a full understanding of the ALAD catalytic mechanism. In addition, given the high enzymatic activity of $[Fe_4S_4]$-ALAD, the condensation reaction of two ALA molecules into porphobilinogen catalyzed by the enzyme may already have been completed by the time the sample was frozen for Mössbauer and EPR analyses, which would leave the $[Fe_4S_4]$-ALAD in the same $[Fe_4S_4]$ coordination status as in the absence of the substrate. Finally, it is possible that the $[Fe_4S_4]$ cluster of ALAD plays solely a structural role rather than a catalytic role. Thus, the role of the $[Fe_4S_4]$ cluster in the enzymatic mechanism of ALAD remains an open question that will require insights into the role of octamerization and cooperativity of individual monomers for further analysis.

Most Fe-S proteins coordinate their clusters through cysteines[17,69]. To identify the cysteines important for $[Fe_4S_4]$ cluster acquisition and/or coordination, we mutagenized the six highly conserved cysteines of human ALAD to alanine residues (A). According to previous studies, lysine 263 (K263) in yeast ALAD, which corresponds to K252 in human ALAD, forms Schiff base with one of the two molecules of ALA to facilitate the condensation reaction[47,70]. Therefore, we also mutagenized K252 in human ALAD to methionine to investigate the impact of a K252M mutant on $[Fe_4S_4]$ cluster coordination and ALAD enzymatic activity. To examine the biological significance of the AFR motif, we substituted the F and R residues with alanines to generate Mut AFR-AAA. According to our results, all the amino acid residues investigated are important for $[Fe_4S_4]$ cluster coordination and the enzymatic activity of human ALAD in vitro. Our findings are consistent with those of a previous study in which a combination of C122A, C124A and C132A greatly reduced activity of human ALAD that ligates zinc[45]. In both the zinc-containing form of the enzyme which can be generated in vitro, and the Fe-S-containing form detected after overexpression in bacteria, cysteines 122, 124, and 132 appear to be required for acquisition of enzymatic activity. Consistent with these findings, a Cys132Arg mutation was found to be responsible

for ALAD porphyria in a 14-year-old boy[71]. Furthermore, we speculate that the structure of ALAD with a $[Fe_4S_4]$ cluster may resemble the crystal structure of Zn-ALAD (Supplementary Fig. 7)[46], and Cys residues 122, 124, and 132, the Zn(II) ligands in the published structures of ALAD[46], might coordinate the $[Fe_4S_4]$ cofactor, potentially leaving an open Fe ligand site[40,60,63]. Cys119 and Cys162 are located on two parallel beta-strands (Supplementary Fig. 7) in positions where they have been proposed to form a disulfide bond that stabilizes ALAD tertiary structure[45]. Therefore, mutagenesis of the cysteine residues that ligate zinc in the available crystal structure may directly affect ISC ligation, whereas mutagenesis of C119 and C162 can impair cluster ligation as a result of misfolding of ALAD. The negative effects on ALAD activity and Fe-S cluster coordination caused by mutagenesis of C223 can be explained on the basis of published structure of the zinc-binding form of the enzyme, in which C223 protrudes at the interface between adjacent monomers, likely contributing to ALAD oligomerization by contacting residues on the adjacent protomer[44]. An alternative possibility is that C223 may represent the fourth cysteine residue ligating the Fe-S cluster, given its proximity to C122, C124, and C132 (Supplementary Fig. 7). Notably, the K252M mutant completely lost its enzymatic activity, as expected based on the observation that it forms a Schiff base to bind ALA substrate[47,70]. We speculate that binding of one of the two ALA molecules, which are condensed to form porphobilinogen in the "active site pocket" as previously defined, was lost[47,70], but Fe-S ligation was also notably absent as well for unclear reasons.

The quaternary structure of human ALAD can switch between a homo-hexameric and a homo-octameric configuration[10]. Wild-type ALAD monomers tend to form a homo-octamer with full enzymatic activity through interactions between their N-terminal domains[10,72]. In the presence of ALAD mutations or of "morphlocks", small molecules that stabilize specific quaternary structures, ALAD can assemble into a homo-hexameric configuration with lower enzymatic activity than the homo-octameric configuration[72,73]. In this study, we found that $[Fe_4S_4]$ cluster coordination correlated with increased proportions of ALAD homo-octamers, which may be another reason why $[Fe_4S_4]$-ALAD has high enzymatic activity.

Finally, we explored the biological significance of the $[Fe_4S_4]$ cluster of ALAD in human HepG2 cells. We performed functional assays to investigate whether overexpression of either wild-type human ALAD or the AFR-AAA mutant could rescue the heme biosynthetic defect of ALAD-knockdown cells. According to our results, the $[Fe_4S_4]$ of human ALAD was important for its enzymatic activity and heme biosynthesis in HepG2 cells. Interestingly, we also found that increased intracellular zinc concentrations, achieved by adding $ZnCl_2$ into the cell culture medium, did not increase ALAD enzymatic activity in wild-type ALAD- or AFR-AAA mutant-overexpressing HepG2 cells, suggesting that mammalian cells lack mechanisms that specifically deliver zinc to ALAD in growing cells.

In conclusion, our findings have several important biological implications. In iron-deficient non-erythroid cells, reduced ISC availability should decrease the enzymatic activity of ALAD, diminishing the potential accumulation of pyrroles and porphyrins synthesized downstream of ALAD that could otherwise occur in conjunction with reduced holo-FECH levels and/or increased ALAS1 protein levels. Porphyrin intermediates are toxic, and their accumulation causes several types of human porphyria[74–76]. Therefore, the regulation of ALAD activity by ISC biogenesis may represent a mechanism that prevents iron- or ISC-deficient non-erythroid cells from accumulating harmful heme metabolic intermediates. In addition, given that Fe-S clusters are sensitive to oxidative stress[17] and that ALAD was shown to be a direct target of oxidative stress[77], we speculate that oxidative damage of the $[Fe_4S_4]$ of human ALAD may be implicated in pathological conditions associated with both increased oxidative stress and anemia, such as chronic renal failure[78]. Furthermore, since we confirmed the biological significance of a LYR or LYR-like motif, our study supports our model that enzymes unrelated to the mitochondrial respiratory chain depend on Fe-S delivery from a cochaperone/chaperone pathway that is critical for Fe-S acquisition[28]. Our work validates our previous prediction[30] that novel Fe-S proteins may be identified prospectively based on presence of LYR motifs in proteins that contain enough cysteine ligands to coordinate a Fe-S cluster, particularly if the protein has been previously characterized as a zinc-binding protein[30].

## Methods

**Plasmids and mutagenesis.** Plasmids used in this study were customized and purchased from GenScript. The mutations in human *ALAD* gene were introduced using the Phusion Site-Directed Mutagenesis Kit (ThermoFisher Scientific) with polymerase chain reaction (PCR) primers listed in Supplementary Table 1 of the Supplementary Information (SI) file.

**Cell lines and culturing methods.** In this study, we used different cell lines for different experiments because heme biosynthesis is a ubiquitous process occurring in all the cell lines and each cell line has characteristics that allow it to perform better in the corresponding experiment. Also, since there is only one known form of endogenous ALAD in all the cell lines used and that we sequenced all the ALAD-expressing plasmids to make sure they express the correct protein, we confirm that we are working with the same protein, human ALAD, in all of our experiments. All the cell lines used in this study were subjected to mycoplasma testing.

We used HeLa cells (ATCC) for investigating the endogenous interaction between HSC20 and ALAD and the interactions between endogenous HSC20 and overexpressed FLAG/MYC-tagged wild-type ALAD or FLAG/MYC-tagged AFR-AAA ALAD mutant. For investigating whether FECH was the only iron-sulfur protein involved in heme biosynthesis, we used HepG2 cells (ATCC) and previously established HEK293T cell lines stably transfected with pLVX-TetOne empty vector, or with pLVX-TetOne-ISCU1-MYC or pLVX-TetOne-ISCU1-D46A mutant-MYC, which expressed MYC-tagged wild-type cytosolic ISCU (ISCU1) or the dominant negative mutant ISCU1-D46A, respectively, under the control of a doxycycline inducible promote[23]. Transgene expression was induced by addition of 2 µg/mL of doxycycline to the growth medium, and cells were harvested and extracted 24 h later. To investigate the biological significance of the $[Fe_4S_4]$ cluster of ALAD, we used HepG2 cells. To evaluate $^{55}Fe$ incorporation into C-terminally FLAG/MYC-tagged POLD1, ALAD WT, and Mut AFR-AAA, we used HEK293 cells (ATCC). To investigate whether human ALAD can coordinate an iron-sulfur cluster in human cells, we overexpressed C-terminally FLAG/MYC-tagged wild-type ALAD and FLAG/MYC-tagged AFR-AAA ALAD mutant in Expi293 cells (ThermoFisher Scientific). The HeLa and HepG2 cells were maintained in Eagle's minimum essential medium (EMEM) supplemented with 10% fetal bovine serum (Gibco, USA) and penicillin-streptomycin (Gibco, USA). The stable inducible HEK293T cells expressing ISCU1 wild-type or mutant were maintained in Dulbecco's Modified Eagle Medium (DMEM), supplemented with 10% of tetracycline-free FBS (Clontech), 2 mM glutamine and 0.25 µg/mL of puromycin. The HEK293 cells were propagated in Dulbecco's modified Eagle's medium (DMEM) with 4.5 g/L glucose, supplemented with 10% fetal bovine serum (FBS) and 2 mM glutamine. The Expi293 cells were propagated and transfected according to the manufacturer's instructions. All the cells were cultured at 37 °C, 5% $CO_2$ in a humidified cell culture incubator.

**Cell transfection and zinc treatment.** HeLa cells were seeded at ~$4 \times 10^6$ cells per 10-cm cell culture plate one day before transfection. The next day, cells were transfected with pCMV6-ALAD wild-type-FLAG/MYC, which directs expression of C-terminally FLAG/MYC-tagged wild-type ALAD protein, or pCMV6-ALAD-Mut AFR-AAA-FLAG/MYC, which directs expression of C-terminally FLAG/MYC-tagged AFR-AAA ALAD mutant, using the Lipofectamine® 2000 Reagent (Invitrogen) according to the protocol provided by the manufacturer. The cells were harvested for downstream analyses ~48 h after transfection.

HepG2 cells were seeded at ~$3 \times 10^6$ cells per 10-cm cell culture plate one day before transfection. The next day, cells were transfected with the ALAD siRNA pool (Dharmacon), the ISCU siRNA pool (Dharmacon), or the combinations of ALAD siRNA+ pCMV6-ALAD wild-type-FLAG/MYC, ALAD siRNA+ pCMV6-ALAD-Mut AFR-AAA-FLAG/MYC, or ISCU siRNA+ pCMV6-yeast FECH-FLAG/MYC (which directs expression of C-terminally FLAG/MYC-tagged yeast FECH). HepG2 cells transfected with a nontargeting siRNA or the pCMV6-Entry empty vector were the negative controls. The transfected HepG2 cells were cultured for 48 h before they were subjected to a second round of transfections with the

same siRNAs, plasmids or siRNA+plasmid combinations. All the transfections were performed using the Lipofectamine® RNAiMAX Reagent (Invitrogen) according to the manufacturer's instructions.

On-TARGETPlus siRNA pools against human *ISCU* (L-012837-01-0005) and the control nontargeting pool (D-001810-10-05) were purchased from Dharmacon. Knockdown of ISCU in HEK293 cells was achieved by transfecting the cells twice with siRNAs at a 48-h interval using Dharmafect 1 according to manufacturer's instructions. Cell lysates were analyzed 48 h after the second transfection.

To test the effect of zinc addition on ALAD activity, 100 μM ZnCl$_2$ was added to the HepG2 cells transfected with the combinations of ALAD siRNA+ pCMV6-ALAD wild-type-FLAG/MYC and ALAD siRNA+ pCMV6-ALAD-Mut AFR-AAA-FLAG/MYC 24 h after the first round of transfections. The cells were collected 72 h after the first round of transfections.

**Cell fractionation and anaerobic purification of recombinant wild-type and mutant ALAD from mammalian cells**. Organelle and cytosolic fractions were separated with the Mitochondria Isolation Kit (ThermoFisher) according to the manufacturer's protocol.

Immunoprecipitations of recombinant FLAG/MYC-tagged POLD1 (POLD1), wild-type ALAD (ALAD WT), or ALAD mutant of the AFR motif (Mut AFR-AAA) were performed on cytosolic fractions from HEK293 or Expi293 cells in a nitrogen recirculated glove box operated at <0.2 ppm O$_2$ 48 h after transfection with plasmids encoding POLD1, ALAD WT, or Mut AFR-AAA. Anti-FLAG immunoprecipitations were performed using M2-FLAG beads (Sigma). Washed FLAG M2 beads were added to the lysate and incubated for 2 h at room temperature under mild agitation. After the beads were recovered, recombinant proteins were competitively eluted with 100 μg/ml 3 × FLAG peptide (Sigma).

UV-vis spectra were acquired using a NanoDrop spectrophotometer (ThermoFisher) using a column buffer supplemented with 100 μg/ml 3 × FLAG peptide as the blank.

**$^{55}$Fe incorporation assay**. HEK293 cells were grown in the presence of 1 μM $^{55}$Fe-Tf and transfected twice at 48 h interval with siRNAs targeting ISCU or with nontargeting siRNAs. At the time of the second transfection with siRNAs, cells were cotransfected with FLAG/Myc-tagged ALAD WT or Mut AFR-AAA, or with FLAG/Myc-tagged POLD1, which was included in the experiment as a positive control. Cytosolic extracts were prepared in the glove box after the second transfection and subjected to immunoprecipitation with anti-FLAG M2 agarose beads.

$^{55}$Fe incorporation into recombinant proteins was measured by scintillation counting of FLAG beads after immunoabsorption of FLAG-tagged recombinant proteins, followed by extensive washings with buffer I [25 mM Tris, 0.15 M NaCl, 1 mM EDTA, 1% NP-40, 5% glycerol (pH 7.4), protease and phosphatase inhibitor cocktail with no EDTA (Roche)]. The background, corresponding to $^{55}$Fe measurements of eluates after anti-FLAG immunoprecipitations on cytosolic extracts from cells transfected with the empty vector (pCMV6-Entry; Origene), was subtracted from each reading.

**Overexpression and anaerobic purification of maltose-binding protein (MBP)-tagged human ALAD proteins**. BL21(DE3) competent bacterial cells (NEB) were cotransformed with pDB1282, a plasmid directing expression of the *isc* operon from *Azotobacter vinelandii*, which consists of bacterial IscS (the bacterial homolog of human NFS1), IscU (the bacterial homolog of ISCU) and IscA (both of which provide scaffolds for iron-sulfur cluster assembly), Fdx (which plays the role of human FDXs by providing reducing equivalents), and HscA and HscB (which have functions corresponding to those of the human chaperone-cochaperone pair, HSPA9 and HSC20, respectively), and pMALc5x-MBP-ALAD plasmids, which directs expression of N-terminally MBP-tagged wild-type or mutant human ALAD proteins. The cotransformants were then used to inoculate 1 L Terrific Broth (TB) medium supplemented with 0.2% glucose and 1% glycerol. After growth at 37 °C for 3–4 h at an optical density at 600 nm (OD$_{600}$) of 0.5–0.7, the culture was cooled to ~15 °C. It was then supplemented with 100 μM IPTG, 0.2% L-(+)-arabinose (Sigma), 300 μM L-cysteine and 40 μM FeCl$_3$ or $^{57}$FeCl$_3$ (for Mössbauer analysis) and cultured at 15 °C for 16–18 h with shaking at 140 rpm. The cells were harvested by centrifugation at 14,000 × *g* and 4 °C and were lysed with the BugBuster® Protein Extraction Reagent (Novagen) supplemented with the rLysozyme™ Solution (Sigma Millipore), Benzonase® Nuclease (Sigma Millipore) and cOmplete™, EDTA-free Protease Inhibitor (Sigma Millipore) in a nitrogen recirculated glove box operated at <0.2 ppm O$_2$. The N-terminally MBP-tagged ALAD proteins in the bacterial lysates were captured by amylose resin (NEB) and eluted with 10 mM maltose in column buffer (20 mM Tris-HCl, pH 8.0, and 200 mM NaCl in deionized water).

**Ultraviolet-visible (UV-vis) spectroscopy, inductively coupled plasma mass spectrometry (ICP-MS), amino acid analysis (AAA), electron paramagnetic resonance (EPR) spectroscopy, and Mössbauer spectroscopy**. Uv-vis spectra were acquired using a NanoDrop spectrophotometer (ThermoFisher) with column buffer supplemented with 10 mM maltose being used as the blank. For ICP-MS, 25 μL sample was digested overnight and iron and zinc concentrations were determined with an Agilent 7900 ICP-MS system. AAA was performed by Alphalyse to precisely quantify the purified ALAD proteins. To verify the presence

of an iron-sulfur cluster, the anaerobically purified MBP-ALAD (250 μM) was subjected to both EPR and Mössbauer analyses.

To verify the presence of iron-sulfur cluster(s), Mössbauer and EPR analyses were carried out. For Mössbauer analysis, a 330 μL aliquot of anaerobically purified, $^{57}$Fe enriched, wild-type (wt) ALAD (250 μM) was transferred into a Mössbauer cup and frozen by placing it on an aluminum block precooled in LN$_2$. In order to assess if binding of the substrate (ALA) or the inhibitor (SA) perturbs the Mössbauer spectrum of the [Fe$_4$S$_4$]$^{2+}$ clusters, two aliquots (330-μL each) of $^{57}$Fe-wt-ALAD (250 μM) were mixed with 33 μL of 100 mM ALA and SA, respectively. The reaction mixtures were subsequently transferred into two Mössbauer cups and frozen as already described. The reaction time between the time of addition of substrates to the time of freezing is ~4 min. Mössbauer spectra were recorded on an alternating constant acceleration Mössbauer spectrometer from SEE Co. (Edina, MN) equipped with a Janis SVT-400 variable-temperature cryostat. The external 53 mT magnetic field was oriented parallel to the direction of propagation of the γ beam. All isomer shifts are quoted relative to the centroid of the spectrum of α-iron metal at room temperature. Simulation of the Mössbauer spectra was carried out using the WMOSS spectral analysis software (www.wmoss.org; Seeco Research, Edina, MN).

For EPR analysis, a 150 μL aliquot of anaerobically purified wild-type ALAD (250 μM) was transferred into an EPR tube and frozen in liquid nitrogen (LN$_2$). To test if ALA or SA binding causes any perturbation to the one electron reduced [Fe$_4$S$_4$]$^+$ cluster in ALAD, we prepared three parallel samples with a final ALAD concentration of 150 μM. Na$_2$S$_2$O$_4$ (final concentration = 2.5 mM) was added to all the samples and incubated for 5 mins. To two of the three samples were added either ALA or SA (final concentrations = 10 mM) before transferring them into two separate EPR tubes and being frozen in LN$_2$. The reaction time of the dithionite-reduced ALAD with substrates (ALA or SA), from the time of mixing to the time of being frozen, is ~3 min. The continuous-wave EPR (CW-EPR) spectrum was acquired on an ESP300 Bruker X-band spectrometer equipped with an ER 410ST resonator. The temperature was controlled by an ER 4112-HV Oxford Instruments (Concord, MA) variable-temperature helium flow cryostat. Acquisition and control of experimental parameters were performed using EWIN 2012 software on an external personal computer via a GPIB interface. The conversion time and the time constant were 40.96 ms; the modulation amplitude was 5 G; the microwave frequency was 9.621 GHz; and the microwave power was 20 mW. Spectra were averaged over 10 scans.

**Preparation of apo-ALAD and zinc-bound ALAD**. Aerobically purified human ALAD was treated with 10 mM EDTA for one hour and passed through a G-25 column (GE Healthcare) to yield metal-free ALAD (apo-ALAD). For zinc reconstitution, apo-ALAD was incubated with 5 mM dithiothreitol and 20 equivalents of ZnCl$_2$ at room temperature for two hours and subsequently passed through a G-25 column.

**Western blotting, co-immunoprecipitation, and native gel electrophoresis**. Organelle and cytosolic fractions were lysed with M-PER® Protein Extraction Reagent (ThermoFisher). For western blotting, 20–40 μg protein sample was used. Mouse monoclonal anti-FLAG antibody was from OriGene, anti-ISCU rabbit polyclonal serum was raised against a synthetic peptide fragment, as previously reported[25]; anti α-TUB antibody was from Sigma. All the other primary antibodies for western blotting were from Abcam.

To investigate the endogenous interaction between human ALAD and HSC20, rabbit anti-HSC20 antibody (Sigma) was coupled onto the amine-reactive resin. To investigate the interactions between endogenous HSC20 and overexpressed FLAG/MYC-tagged wild-type ALAD or FLAG/MYC-tagged AFR-AAA ALAD mutant, EZview™ Red ANTI-FLAG® M2 Affinity Gel (Sigma) was used. Cytosolic fractions and whole-cell lysates of HeLa cells were incubated with the anti-HSC20 antibody-coupled resin or the anti-FLAG beads at 4 °C overnight, and bound proteins were collected by acidic elution (Tris-glycine, pH 2.8). The eluates were then subjected to western blotting with anti-HSC20 and anti-ALAD antibodies (Abcam). Co-immunoprecipitation with rabbit IgG served as a negative control, and aliquots corresponding to 5–10% of HeLa cytosolic fractions or whole-cell lysates were used as inputs.

For native gel electrophoresis, anaerobically purified ALAD proteins were separated, and the bands were visualized using the NativePAGE™ Bis-Tris Gel System (ThermoFisher) according to the manufacturer's instructions.

**Antibody information**. In this study, we used the following antibodies for Western blotting and co-immunoprecipitation experiments: rabbit anti-ALAD (1:1000, Abcam ab59013); rabbit anti-ALAS1 (1:500, Abcam ab84962); rabbit anti-FECH (1:500, Abcam ab137042); mouse anti-TOMM20, (1:2000, Abcam ab56783); mouse anti-β-Tubulin, (1:2000, Sigma T8328); mouse anti-α-Tubulin, (1:2000, Sigma T5168); rabbit anti-HSC20, (1:50 for Co-IP experiments, Sigma HPA018447); mouse anti-HSC20, (1:1000, OriGene TA507285); mouse anti-FLAG Tag, (1:100 for Co-IP experiments, Genscript A00187); rabbit anti-MT2A, (1:200, LifeSpan Bio LS-C667906-50); mouse anti-FLAG, (1:2000, OriGene TA50011-100); rabbit anti-ISCU antibody, (1;1000, prepared in our lab).

**ALAD, cytochrome P450 (CYP), and catalase activity assays**. ALAD activity was determined according to a published protocol[79]. Briefly, 50 µl (1 µg/µl) of purified ALAD proteins or freshly prepared cell lysates were added into a reaction system containing 238 µl of ice-cold Tris-acetate buffer (100 mM, pH 7.2) and 12 µl of 100 mM ALA-HCl. The mixtures were incubated at 37 °C for 10 min (for purified ALAD proteins) or one hour (for cell lysates). The reactions were terminated by adding 300 µl of 10% trichloroacetic acid/0.1 M $HgCl_2$ and the amounts of generated porphobilinogen were quantified using the modified Ehrlich's reagent. The CYP1A2 and catalase activities were analyzed by the CYP1A2 Activity Assay Kit (Abcam) and Catalase Activity Assay Kit (Abcam), respectively.

**Statistics**. Image J v1.46 was used to analyze band intensity of western blotting. Where applicable, three independent replicates (four independent replicates for the [55]Fe incorporation experiments) were performed, and results are presented as mean ± standard deviation (S.D.) using the Microsoft Excel 2016 Two-sided unpaired Student's $t$-test was used to determine the P value using GraphPad Prism 8. $P < 0.05$, $P < 0.01$, and $P < 0.001$ were considered significant (* or [a]), very significant (** or [aa]) and extremely significant (*** or [aaa]), respectively.

**Reporting summary**. Further information on research design is available in the Nature Research Reporting Summary linked to this article.

## Data availability

All the data reported in this study are available upon reasonable request. Source data are provided with this paper.

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

## Acknowledgements
We are grateful to all the members of our lab for constructive discussions and to Dr. Gregory Holmes-Hampton for his help on ICP-MS analysis. This work was supported by the Intramural Program of the *Eunice Kennedy Shriver* National Institute of Child Health and Human Development and the National Institutes of Health (GM-127079 to C.K.).

## Author contributions
G.L. designed and performed the experiments, analyzed the data, and wrote the manuscript. D.S. performed the Mössbauer and EPR experiments, analyzed the data, and wrote the manuscript. N.M. designed and performed experiments and analyzed the data. W.T. analyzed the data. J.M.B. and C.K. analyzed the Mössbauer and EPR data. T.A.R. designed and supervised the study, analyzed the data, and wrote the manuscript. All authors revised the article critically.

## Competing interests
The authors declare no competing interests.
