## [Peer Review File · Nature Communications]

REVIEWER COMMENTS

Reviewer #1 (Remarks to the Author):

The authors have done an excellent job of addressing the major issue outlined by this reviewer by providing concrete evidence for Fe-S incorporation into ALAD in human cells. However, a few minor issues remain that will require minor edits:

1) Fig. 2 caption: The descriptions of panels A and C should be swapped to match the figure and the main text.

2) Fig. 2D: The authors are asked again to include the Zn form of WT ALAD in this native PAGE analysis as an additional control since this form was also assessed for ALAD activity. Does the higher activity of Zn-ALAD compared to the Cys mutants correlate with increased octamer levels? Native PAGE of the Zn form has been published previously as the authors pointed out in their Response to Referees; however, it would be instructive to observe whether the protein behaves the same via this type of analysis in their hands.

Reviewer #3 (Remarks to the Author):

The most essential criticism for the original version of the manuscript was the lack of the experimental evidence of the formation of FeS cofactors without overexpressing the bacterial ISC operon in mammalian cells. The additional experiments as shown in Fig. 1 clearly confirm the presence of the FeS cluster in mammalian cells under the physiological conditions. Congratulations! A small comment is that the reasons why the authors used different cell lines for ⁵⁵Fe incorporation assay (HEK293), purification of recombinant ALAD (Ecp1293) and enzymatic activity (HepG2) should be added. Some description should also be added to confirm that the same form of ALAD is expressed in these cells.

The reviewer is still concerned that the spectroscopic characterization of the [Fe₄S₄] cluster in ALAD is incomplete, but also understands the difficulty of the sample preparation and the measurements without minor components. Measurements of absorption spectra for ALAD from mammalian cells

would also require laborious sample preparation, particularly under the situation of the pandemic. The reviewer accepts the claim of the authors that they added a new insight into the cofactor of ALAD and they cannot solve all the remaining mechanistic questions in this paper.

The response to the comment on the results of the immunoprecipitation is reasonable and the short discussion on the detection of the transient interactions between ALAD and HSC20 should be added to avoid misunderstanding for general readers.

Although the authors responded to the comment on the activity of the Cys-mutated ALAD mutants, but no description of the C223 mutant was added to the revised version of the manuscript. Some (but short) discussion is required for the low activity of the Cys mutant.

Overall, providing the experimental evidence for the formation of the FeS cluster in mammalian cells under physiological conditions is quite essential for the publication, which is satisfied in the revised version of the manuscript, although some spectroscopic characterization would not be complete. After some minor revisions, the reviewer recommends that the manuscript can be acceptable for the publication.

Review of Liu et al.

Comment 1: In my opinion, the manuscript should be accepted for publication in Nature Communications. In it, the authors present strong evidence that under normal cellular conditions, ALAD contains an [4Fe4S] cluster and that this is the physiologically relevant form rather than Zn-bound. This is an exciting and important result. I have come to this decision based on the science presented, including my evaluation of the extensive responses of the authors. That being said, I do have some final comments and suggestions.

Comment 2: Another reviewer and I asked the authors to include the Zn form of WT ALAD in the native PAGE analysis. The response was that doing so would be excessively difficult; however, purifying the enzyme in air and adding a Zn salt does not seem difficult and in fact they report doing this in other parts of the paper. I still think it would be a good addition.

Comment 3: The authors agreed with me that the Zn and Fe centers “likely serve the same structural role” rather than a catalytic role, yet they continue to assume a catalytic role in their paper. I’m confused by this.

Comment 4: They authors agreed that the EPR and Mossbauer studies were incomplete, but did nothing to improve this section apart from moving the EPR spectrum from the main text to SI. I never asked for any advanced spectroscopic studies (ENDOR, ESEEM, HYSCORE) or for rapid-freeze-quench kinetic studies to be carried out - only that they “add substrate and see if one of the cluster irons is affected” and “reduce the sample, with and without substrate bound” and examine whether they could detect substrate binding using EPR and Mossbauer. I regard this as a reasonable request since the authors hypothesize in the paper that the cluster has an open site that would allow for substrate binding. Moreover, two of the authors are extraordinarily capable of performing such studies, and it would certainly strengthen the paper and their claim.

Comment 5: The authors agreed that the presence of an Fe/S cluster in mammalian ferrochelatase previously explained how heme biosynthesis is regulated/connected to iron-sulfur biosynthesis. So finding another Fe/S-containing protein in the heme biosynthesis pathway (in mammals) adds to this explanation as there now appears to be (at least) two sites of regulation. However, the authors continue to push the idea that the discovery that ALAD is the sole (or dominant) explanation for the connection between Fe/S and heme biosynthesis – including in the title. I agree that showing an Fe/S cluster in yeast ALAD would show this as the dominant regulatory site, since yeast ferrochelatase does not contain an Fe/S cluster - but their article focuses on the human enzyme, not yeast. Clarifying this for readers who are unfamiliar with these details would be helpful, but doing so would require toning-down their conclusions and perhaps modifying the title.

Comment 6: Regarding the LYR motif, the authors may be correct to focus on this motif, but they should also consider the following analysis. The probability of generating this sequence randomly is

$$P_{LYR} = \frac{1}{20} \cdot \frac{1}{20} \cdot \frac{1}{20} = \frac{1}{8000}$$

Thus, the occurrence of this sequence has a fairly low probability. However, for the first position, there are 6 letters that “count” (L, T, M, P, V, I), for the second position, there are 2 letters that ‘count’ (Y or F), and for the third position, there are 2 letters that “count” (R or K). The probability of finding any one of these “LYF-like” sequences in a protein would be 24 times higher...

$$P_{LYR \text{ or } LYR-LIKE} = 6 \cdot 2 \cdot 2 \cdot \frac{1}{8000} = \frac{24}{8000} = 0.003$$

This means that there will be a LYR-like motif every 333 “three-amino-acid sequences”. Consider a protein with a molecular weight of 33,000 Da (300 amino acid sequences). If we divided the 300 amino acid sequences into groups of three, the probability of there being an LYR-like motif would be

$$P_{LYR \text{ or } LYR-LIKE \text{ in an average protein}} = 100 \cdot \frac{24}{8000} = \frac{2400}{8000} = 0.3$$

This means that by random occurrence, there will be an LYR-like motif in about 30% of all proteins. Actually by considering each of the 300 amino acids in the protein as the start of a three-amino-acid sequence, the likelihood of there being an LYR-like motif starting at any of these 300 amino acid positions would be 100% (i.e. every average-length protein would have one LYR-like sequence). Of course amino acid sequences are not random, so the actual likelihood could either be lower or higher. But my point is that discovering an LYR-like sequence in a protein does not automatically or even strongly indicate that it is used as a signal for installing an Fe/S cluster; it could simply be a random occurrence. The fundamental problem, from a statistical perspective, is that the sequence is too short and there are too many “options” for what counts as membership in the LYR group. A longer sequence with fewer options would be more definitive.

RESPONSES TO REVIEWERS' COMMENTS

Reviewer #1 (Remarks to the Author):

The authors have done an excellent job of addressing the major issue outlined by this reviewer by providing concrete evidence for Fe-S incorporation into ALAD in human cells. However, a few minor issues remain that will require minor edits:

1) Fig. 2 caption: The descriptions of panels A and C should be swapped to match the figure and the main text.

Response: *We have corrected the order of description of panels A, B and C in Fig. 2 caption.*

2) Fig. 2D: The authors are asked again to include the Zn form of WT ALAD in this native PAGE analysis as an additional control since this form was also assessed for ALAD activity. Does the higher activity of Zn-ALAD compared to the Cys mutants correlate with increased octamer levels? Native PAGE of the Zn form has been published previously as the authors pointed out in their Response to Referees; however, it would be instructive to observe whether the protein behaves the same via this type of analysis in their hands.

Response: *We have incorporated the Native PAGE of the aero-Zn-ALAD WT protein into Figure 2D and Figure S7 of our revised manuscript. Our results show that, in contrast to the ALAD Cys mutants that assembled into hexamers, the zinc-bound form of ALAD octamerized, consistent with previously published findings. A full gel with the octamerized zinc form is shown as Figure S7, whereas the Figure 2D includes a splice from this figure.*

Reviewer #2 (Remarks to the Author):

Review of Liu et al.

Comment 1: In my opinion, the manuscript should be accepted for publication in Nature Communications. In it, the authors present strong evidence that under normal cellular conditions, ALAD contains an [4Fe4S] cluster and that this is the physiologically relevant form rather than Zn-bound. This is an exciting and important result. I have come to this decision based on the science presented, including my evaluation of the extensive responses of the authors. That being said, I do have some final comments and suggestions.

Comment 2: Another reviewer and I asked the authors to include the Zn form of WT ALAD in the native PAGE analysis. The response was that doing so would be excessively difficult; however, purifying the enzyme in air and adding a Zn salt does not seem difficult and in fact they report doing this in other parts of the paper. I still think it would be a good addition.

Response: *We have incorporated the Native PAGE of the aero-Zn-ALAD WT protein in the new Figure 2D and Figure S7 in our revised manuscript. A full gel*

with the octamerized zinc form is shown as Figure S7, whereas the Figure 2D includes a splice from this figure.

Comment 3: The authors agreed with me that the Zn and Fe centers “likely serve the same structural role” rather than a catalytic role, yet they continue to assume a catalytic role in their paper. I’m confused by this.

Response: *The reviewer suggests that structural and catalytic roles must be mutually exclusive, but there are examples in which a FeS cluster, by virtue of being ligated by cysteines from distal locations in the primary sequence, drives the fold of the proteins, while at the same time participating in numerous reactions such as electron transfer, as in the case of the B subunit of the succinate dehydrogenase complex (SDHB). Therefore, we speculate that the cluster in ALAD could play both a structural and a catalytic role.*

Comment 4: They authors agreed that the EPR and Mossbauer studies were incomplete, but did nothing to improve this section apart from moving the EPR spectrum from the main text to SI. I never asked for any advanced spectroscopic studies (ENDOR, ESEEM, HYSCORE) or for rapid-freeze-quench kinetic studies to be carried out - only that they “add substrate and see if one of the cluster irons is affected” and “reduce the sample, with and without substrate bound” and examine whether they could detect substrate binding using EPR and Mossbauer. I regard this as a reasonable request since the authors hypothesize in the paper that the cluster has an open site that would allow for substrate binding. Moreover, two of the authors are extraordinarily capable of performing such studies, and it would certainly strengthen the paper and their claim.

Response: *As requested by the reviewer, we prepared a large prep of ⁵⁷Fe-labeled ALAD for additional Mossbauer analysis, performed at Penn State. We found that the Mossbauer spectra were unchanged upon addition of the substrate aminolevulinic acid. EPR signals of ALAD reduced with dithionite were also unchanged upon addition of the substrate. These figures are now included in the paper as new Figures S4 and S5. Accordingly, we have added a last paragraph to the “Human ALAD coordinates a previously unrecognized [Fe₄S₄] cluster” subsection of the Results section and changed the narrative of the 4th paragraph of the Discussion section to better reflect the outcomes of these latest results. We appreciate this good suggestion and were happy that circumstances enabled us to perform the suggested experiments.*

Comment 5: The authors agreed that the presence of an Fe/S cluster in mammalian ferrochelatase previously explained how heme biosynthesis is regulated/connected to iron-sulfur biosynthesis. So finding another Fe/S-containing protein in the heme biosynthesis pathway (in mammals) adds to this explanation as there now appears to be (at least) two sites of regulation. However, the authors continue to push the idea that the discovery that ALAD is

the sole (or dominant) explanation for the connection between Fe/S and heme biosynthesis – including in the title. I agree that showing an Fe/S cluster in yeast ALAD would show this as the dominant regulatory site, since yeast ferrochelatase does not contain an Fe/S cluster - but their article focuses on the human enzyme, not yeast. Clarifying this for readers who are unfamiliar with these details would be helpful, but doing so would require toning-down their conclusions and perhaps modifying the title.

Response: *We have modified the title to: “Human ALAD is an unrecognized iron-sulfur protein that contributes to dependence of heme biosynthesis on iron sulfur cluster biogenesis” as suggested by the reviewer.*

Comment 6: Regarding the LYR motif, the authors may be correct to focus on this motif, but they should also consider the following analysis. The probability of generating this sequence randomly is $\frac{1}{6} \times \frac{1}{2} \times \frac{1}{2} = \frac{1}{24}$. Thus, the occurrence of this sequence has a fairly low probability. However, for the first position, there are 6 letters that “count” (L, T, M, P, V, I), for the second position, there are 2 letters that “count” (Y or F), and for the third position, there are 2 letters that “count” (R or K). The probability of finding any one of these “LYF-like” sequences in a protein would be 24 times higher... $\frac{1}{24} \times 24 = 1$. This means that there will be a LYR-like motif every 333 “three-amino-acid sequences”. Consider a protein with a molecular weight of 33,000 Da (300 amino acid sequences). If we divided the 300 amino acid sequences into groups of three, the probability of there being an LYR-like motif would be $\frac{300}{333} \approx 0.9$ or 90%. This means that by random occurrence, there will be an LYR-like motif in about 90% of all proteins. Actually by considering each of the 300 amino acids in the protein as the start of a three-amino-acid sequence, the likelihood of there being an LYR-like motif starting at any of these 300 amino acid positions would be 100% (i.e. every average-length protein would have one LYR-like sequence). Of course amino acid sequences are not random, so the actual likelihood could either be lower or higher. But my point is that discovering an LYR-like sequence in a protein does not automatically or even strongly indicate that it is used as a signal for installing an Fe/S cluster; it could simply be a random occurrence. The fundamental problem, from a statistical perspective, is that the sequence is too short and there are too many “options” for what counts as membership in the LYR group. A longer sequence with fewer options would be more definitive.

Response: *The reviewer performs calculations based on the possibility that sequences occur randomly in the human genome/proteome. This approach does not take into account many contingencies. The first is that biological processes in eukaryotes occur in a compartment-specific location, such as Fe-S biogenesis that occurs in the mitochondrial matrix and in the cytosolic/nuclear compartments of mammalian cells, but not in the secretory pathway, plasma membrane, vesicles, lysosomes, etc. We agree that this motif*

is not rare and cannot always signify that a Fe-S cofactor will be incorporated by the protein that contains the motif in its primary sequence. The motif has been experimentally demonstrated to be necessary for binding of the FeS biogenesis cochaperone HSC20, but apart from the subcellular colocalization of the cochaperone with the target recipient FeS apoprotein, it is not yet clear how greater discrimination is achieved. Nevertheless, we pursued ALAD on the basis of the motif and of a zinc-binding site (which has been demonstrated to be able to occupy FeS binding sites in proteins purified aerobically), and we performed a thorough investigation that demonstrated that ALAD binds a Fe-S cofactor. For the reviewer's issues about the amount of information that is needed in functional motifs, we recommend an excellent review from Tompa, P., N. E. Davey, T. J. Gibson, and M. M. Babu. 2014. "A million peptide motifs for the molecular biologist". Mol Cell 55: 161-169.

We think that none of these proven motifs would satisfy the reviewer's mathematical predictions, yet the functions are experimentally proven and well-accepted. Several well accepted motifs that were discovered in particular experimental settings have proven to have more generalized relevance. A compendium of motifs has been assembled (<http://elm.eu.org>) together with their functions.

Nevertheless, we have expanded the narrative of the third paragraph of the Introduction section and that of the second paragraph of the Discussion section, and rewritten the last sentence of the Discussion section to address the reviewer's concerns in the revised manuscript.

Reviewer #3 (Remarks to the Author):

The most essential criticism for the original version of the manuscript was the lack of the experimental evidence of the formation of FeS cofactors without overexpressing the bacterial ISC operon in mammalian cells. The additional experiments as shown in Fig. 1 clearly confirm the presence of the FeS cluster in mammalian cells under the physiological conditions. Congratulations!

A small comment is that the reasons why the authors used different cell lines for ⁵⁵Fe incorporation assay (HEK293), purification of recombinant ALAD (Ecp1293) and enzymatic activity (HepG2) should be added. Some description should also be added to confirm that the same form of ALAD is expressed in these cells.

Response: *We would like to thank the reviewer for the nice suggestion. We have added two sentence that read "In this study, we used different cell lines for different experiments because heme biosynthesis is a ubiquitous process occurring in all the cell lines and that each cell line has characteristics that allow it to perform better in the corresponding experiment. Also, since there is only one known form of endogenous ALAD in all the cell lines used and that we sequenced all the ALAD-expressing plasmids to make sure they express the*

correct protein, we confirm that we are working with the same protein, human ALAD, in all of our experiments.” to the Cell culture and transfection methods subsection of the Methods section to address this issue.

The reviewer is still concerned that the spectroscopic characterization of the [Fe₄S₄] cluster in ALAD is incomplete, but also understands the difficulty of the sample preparation and the measurements without minor components. Measurements of absorption spectra for ALAD from mammalian cells would also require laborious sample preparation, particularly under the situation of the pandemic. The reviewer accepts the claim of the authors that they added a new insight into the cofactor of ALAD and they cannot solve all the remaining mechanistic questions in this paper.

Response: *We would like to thank the reviewer for being empathetic! We did pursue more spectroscopic studies and have summarized the results in Figure S4 and Figure S5 of the revised manuscript.*

The response to the comment on the results of the immunoprecipitation is reasonable and the short discussion on the detection of the transient interactions between ALAD and HSC20 should be added to avoid misunderstanding for general readers.

Response: *We have rewritten the penultimate sentence of the second paragraph of the Discussion section to further emphasize that ALAD can transiently interact with HSC20 in human cells.*

Although the authors responded to the comment on the activity of the Cys-mutated ALAD mutants, but no description of the C223 mutant was added to the revised version of the manuscript. Some (but short) discussion is required for the low activity of the Cys mutant.

Response: *We have described the possible function of C233 and how the C223A mutant will affect the function of ALAD protein in the fourth paragraph of the Discussion section of the revised manuscript.*

Overall, providing the experimental evidence for the formation of the FeS cluster in mammalian cells under physiological conditions is quite essential for the publication, which is satisfied in the revised version of the manuscript, although some spectroscopic characterization would not be complete. After some minor revisions, the reviewer recommends that the manuscript can be acceptable for the publication.

REVIEWERS' COMMENTS

Reviewer #2 (Remarks to the Author):

The authors have responded appropriately to all of my previous comments and suggestions. I recommend that the paper be published without further changes.